# Ankyrin repeats in context with human population variation

**Javier S. Utgés**[1,2], **Maxim I. Tsenkov**[1], **Noah J. M. Dietrich**[1], **Stuart A. MacGowan**[1], **Geoffrey J. Barton**[1]*

1 Division of Computational Biology, School of Life Sciences, University of Dundee, Scotland, United Kingdom, 2 Universitat Pompeu Fabra (UPF), Barcelona, Spain

* gjbarton@dundee.ac.uk

**Data Availability Statement:** All data and code relevant to this paper can be accessed on GitHub from: https://github.com/bartongroup/JSU_ANK_analysis and Zenodo: DOI: 10.5281/zenodo.5139587.

## Abstract

Ankyrin protein repeats bind to a wide range of substrates and are one of the most common protein motifs in nature. Here, we collate a high-quality alignment of 7,407 ankyrin repeats and examine for the first time, the distribution of human population variants from large-scale sequencing of healthy individuals across this family. Population variants are not randomly distributed across the genome but are constrained by gene essentiality and function. Accordingly, we interpret the population variants in context with evolutionary constraint and structural features including secondary structure, accessibility and protein-protein interactions across 383 three-dimensional structures of ankyrin repeats. We find five positions that are highly conserved across homologues and also depleted in missense variants within the human population. These positions are significantly enriched in intra-domain contacts and so likely to be key for repeat packing. In contrast, a group of evolutionarily divergent positions are found to be depleted in missense variants in human and significantly enriched in protein-protein interactions. Our analysis also suggests the domain has three, not two surfaces, each with different patterns of enrichment in protein-substrate interactions and missense variants. Our findings will be of interest to those studying or engineering ankyrin-repeat containing proteins as well as those interpreting the significance of disease variants.

## Author summary

Comparison of variation at each position of the amino acid sequence for a protein across different species is a powerful way to identify parts of the protein that are important for its structure and function. Large-scale DNA sequencing of healthy people has recently made it possible to study normal genetic variation within just one species. Our work combines information on genetic differences between over 100,000 people with in-depth analysis of all available three-dimensional structures for Ankyrin repeats, which are a widespread family of binding proteins formed by units with similar amino acid sequence that are found in tandem. Our combined analysis identifies sites critical for ankyrin stability as well as the positions most important for substrate interactions and hence function. Although focused only on the Ankyrins, the principles developed in our work are general and can be applied to any protein family.

**Funding:** GJB and SAM received support from the Biotechnology and Biological Sciences Research Council (BBSRC; https://bbsrc.ukri.org; grant codes: BB/J019364/1 and BB/R014752/1). JSU was supported by a BBSRC EASTBIO Ph.D. Studentship (Grant code: BB/J01446X/1). MIT was supported by a Wellcome Trust Ph.D. studentship (https://wellcome.ac.uk; Grant code: 102132/B/13/Z). GJB also received support from a Wellcome Trust Biomedical Resources Grant (Grant code: 101651/Z/13/Z). The funders had no role in study design, data collection and analysis, decision to publish, or preparation of the manuscript.

**Competing interests:** The authors have declared that no competing interests exist.

## Introduction

The ankyrin repeat motif (ANK) is one of the most commonly observed protein motifs in nature, with proteins containing this motif found in practically all phyla [1]. Ankyrin repeats (AR) are specialised in protein binding and take part in many processes including transcription initiation, cell cycle regulation and cell signalling [2]. ANK is 33 residues long (Fig 1A) and has a helix-turn-helix conformation, with short loops at the N and C termini (Fig 1B). The last and first two residues of adjacent repeats form a β-turn. These β-turns project outward at an angle of ≈ 90˚ to the antiparallel α-helices, yielding the characteristic L-shaped cross section of ankyrin repeats. Ankyrin repeats are usually found in tandem with two or more forming an ankyrin repeat domain (ARD). The stacking of repeats is mediated by the conserved hydrophobic faces of the helices as well as the complementarity of repeat surfaces that assemble to form an extended helical bundle (Fig 1C) [3]. Less conserved positions in the motif, i.e., positions that present residues with many different physicochemical properties, are located on the surface, and are likely to interact with ligands. More conserved positions, on the other hand, tend to be buried in the structure and are responsible for the correct packing of the domain. They do this by forming both intra- and inter- repeat contacts, such as hydrophobic and hydrogen bond interactions [4].

Proteins containing ankyrin repeats are known to bind many different protein and small molecule substrates. The concave face of an ARD, comprising the β-turn/loop region and the first α-helix is often associated with substrate binding. [8]. Recent evidence suggests that ARDs might not only be able to bind small ligands or proteins, but also a range of sugars and lipids, thus extending their versatility and flexibility in substrate binding [9]. This, coupled with the success that designed ankyrin repeat proteins (DARPins) are having in the clinical field, [10] make ankyrins an extremely interesting target to study.

Within the sequence variability found in protein repeats, ankyrins are relatively conserved and multiple amino acid patterns can be observed within the ANK [11] (Fig 1A). The TPLH motif, positions 4–7, is highly conserved across all ankyrin repeats. It is found at the beginning of the first α-helix. Thr4 establishes three hydrogen bonds with His7 (Fig 2E), Pro5 starts the helix with a tight turn and Leu6 forms multiple hydrophobic interactions both within and between repeats (Fig 2B). The loops are more diverse in sequence, yet certain patterns are apparent as well. The subsequence GADVN, 25–29, can be observed in the loop connecting the second α-helix with the β-turn. Gly25 breaks the second helix. Ala26 and Val28 form intra- and inter-repeat interactions (Fig 2B), whereas Asp27 and Asn29 form hydrogen bonds with adjacent repeats (Fig 2D). Gly13 is found in the loop between the antiparallel helices. Asp32 and Gly2 are conserved at the β-turn that connects repeats (Fig 2C). A total of six hydrogen bonds take place in this turn, explaining why Asp32 is conserved. Gly2 performs a similar role to Gly13 and Gly25 and is conserved due to its flexibility and special structural features [12]. In the second α-helix, the [I/V]VXLLL hydrophobic motif is observed. The residues on this motif, except X19, which is usually hydrophilic, form intra- and inter-repeat hydrophobic networks that are thought to help keep together the ARD structure [13] (Fig 2B).

The N- and C- capping repeats are thought to be more flexible than the internal repeats. This is because the capping repeats only have one contact interface, whereas internal repeats have two. Schilling *et al* [15] have demonstrated through molecular dynamics simulations that the region spanning from GADVN to the β-turn, at the interface between adjacent repeats is the most flexible region of the motif. They also proved how the increase in thermostability of the domain, induced by targeted amino acid substitutions, was independent of the binding concave surface, and consequently, of its interaction with the protein substrate. These are the general trends that can be elucidated when doing a comprehensive study of repeats. However,

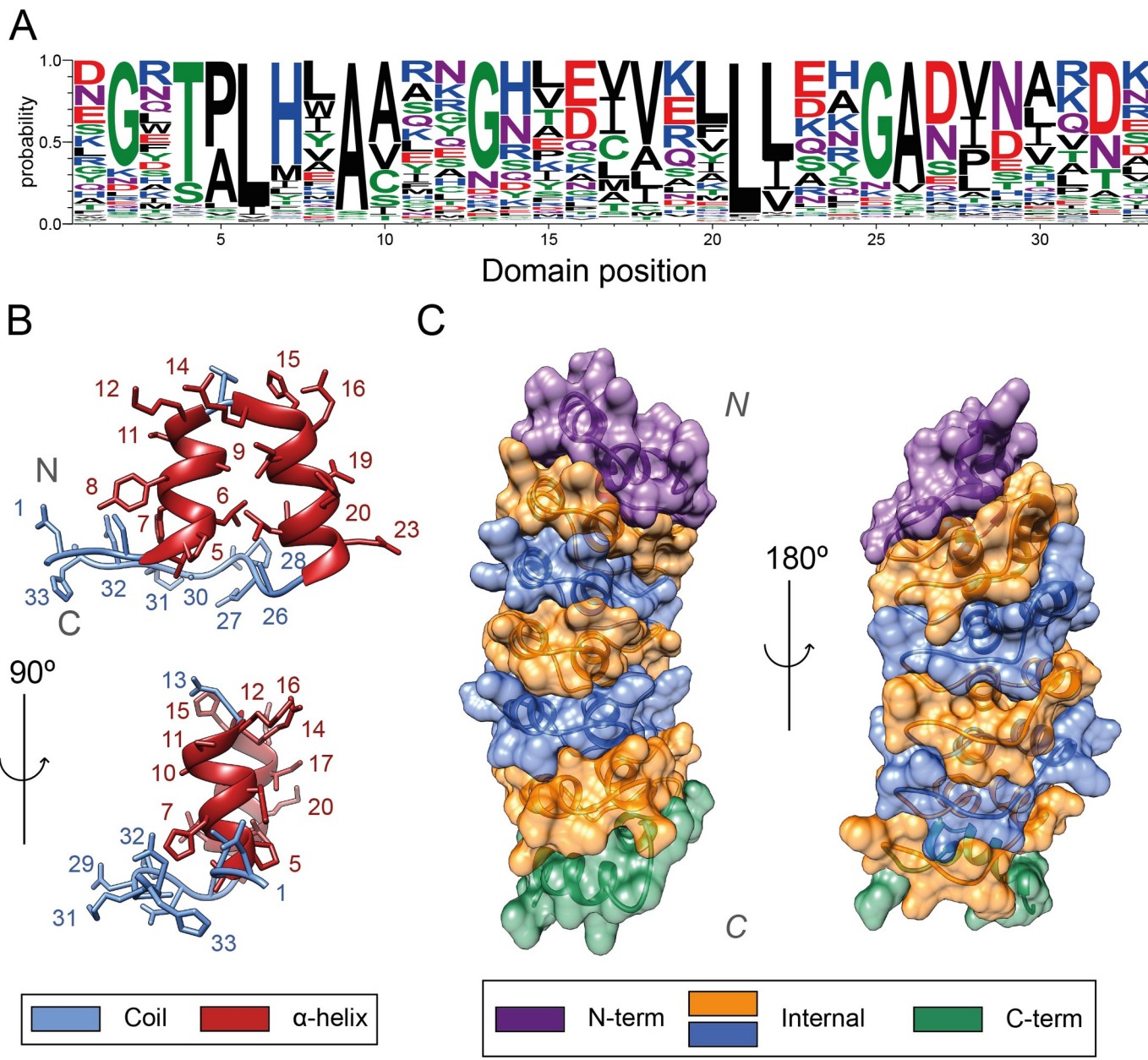

**Fig 1.** (A) Sequence logo of the ANK obtained with WebLogo [5] derived from the MSA generated in this work. The Y axis indicates the probability of observing an amino acid at any position within the motif; (B) Tertiary structure of an ankyrin repeat, coloured by secondary structure class: helices in red and coil in blue; (C) Representation of the complementary surfaces of individual ARs that form the human gankyrin ARD surface. N- and C-capping AR surfaces are coloured in purple and green respectively, whereas internal ones are coloured in blue and orange. (PDB ID: 1UOH) [6]. Structure visualization with UCSF Chimera [7].

there are specific cases where proteins behave differently. Sue *et al* [16] reported how the ankyrin repeats 5 and 6 of IκBα were highly flexible and unstable and this was critical for its function. In fact, substitutions approximating the sequence of these repeats to the consensus ankyrin sequence stabilised the ARD and impaired its function.

In protein sequences, functional or structurally important residues are constrained in evolution, resulting in amino acid conservation between homologues. Conservation amongst all homologues has been exploited to identify important residues, whilst comparing conservation

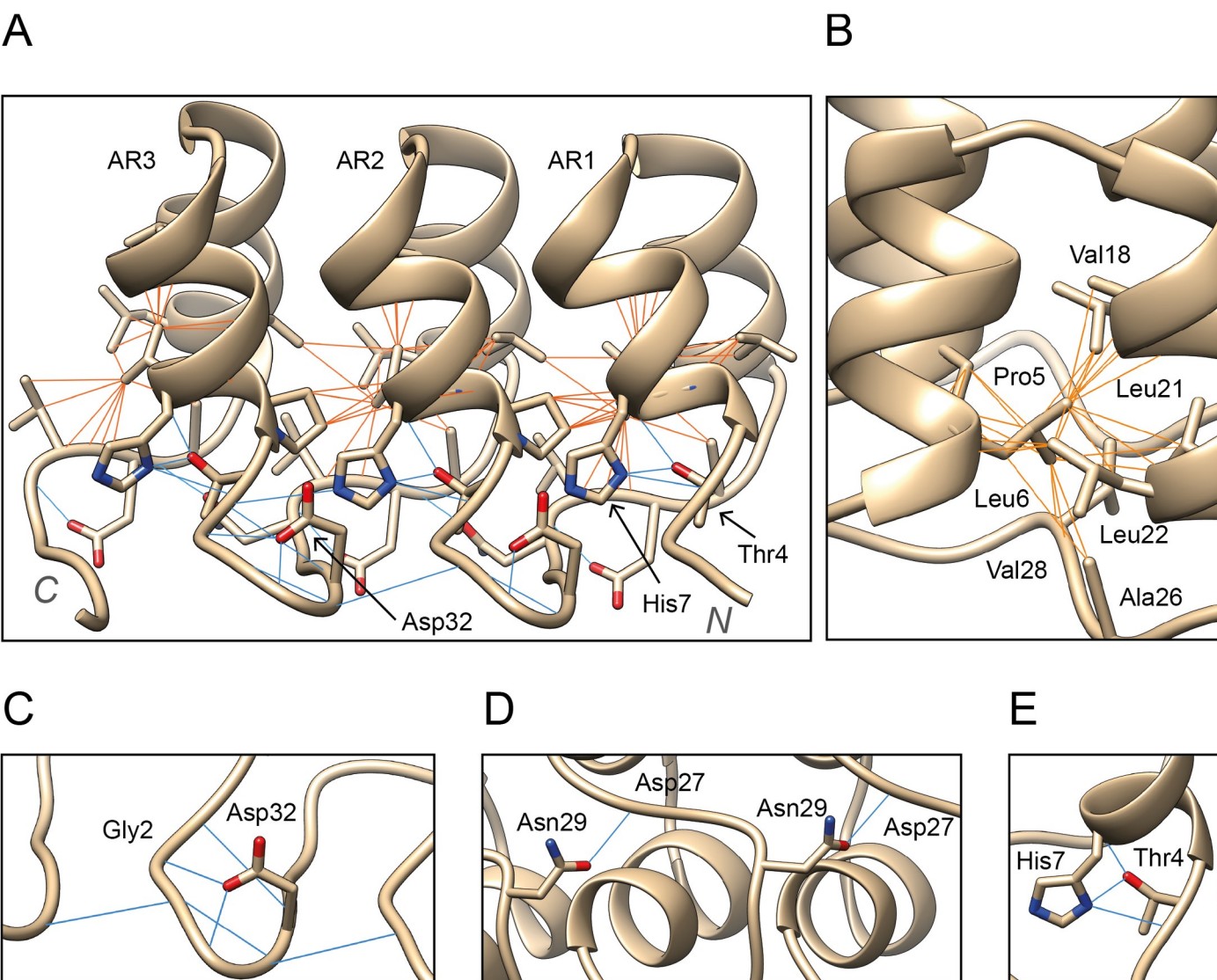

**Fig 2.** (A) Trio of ARs from a designed ankyrin repeat protein [14] (PDB: 5MA3). These three ARs display the main interactions responsible for the correct packing of the ARD. Hydrogen bond interactions are depicted by blue lines whereas hydrophobic ones are depicted in orange; (B) Hydrophobic network formed by Leu6 in the hydrophobic core of the domain; (C) Hydrogen bonding network at the β-turn between positions Asp32-Gly2; (D) Inter-repeat hydrogen bonds between conserved Asn29 and Asp27; (E) Thr4 forms three hydrogen bonds with His7. Structure visualization with UCSF Chimera [7].

between alignments of paralogs and orthologs separately can yield additional insights such as the identification of specificity determining residues [17]. In a similar way to how structure and function constrains the residues within protein families, the distribution of genetic variation within a protein is constrained by domain structure and function [18]. Synonymous variants do not change the protein sequence, and as a result they appear randomly distributed in structure. Missense variants, on the other hand, change the protein sequence and are consequently constrained in space. Pathogenic missense variants cluster in three-dimensional structure around functionally important regions, such as catalytic sites, whereas neutral or non-pathogenic missense variants tend to aggregate on regions which are tolerant to amino acid substitutions [19]. Recently, in a study of all Pfam domain families, MacGowan *et al* [18] found that positions conserved across homologues and also depleted in missense variants

within the human population were of particular functional and/or structural relevance since they are heavily enriched in disease-associated variants in human. They also found a subset of divergent positions across homologues that were missense depleted. These positions were enriched in ligand, DNA and protein interactions as well as in pathogenic variants, suggesting their functional importance within the protein domain.

In this paper we perform a novel analysis that combines human population genetic variation from gnomAD [20] across ankyrin repeats in context with evolutionary variation and all available ankyrin protein structures. This is the first in-depth application to a repeat family of the concepts developed in our earlier work on 1,291 Pfam domains [18]. Genetic variation data are still too sparse to provide a comparative picture between individual residues in the proteome. However, in [18] we overcame this problem by aggregating variants over equivalent positions in multiple sequence alignments of protein homologues. Extending this approach to a repeat family further boosts the statistical power of the method, since repeat domains usually present multiple repeat copies within a protein. The results of this analysis highlight the positions in the ANK most likely to be important for structural stability as well as those relevant to substrate specificity. We anticipate that this work will be of value to those interested in understanding the function of ANK containing proteins as well as those aiming to engineer novel AR specificity.

## Methods

### Sequence extraction and database integration

InterPro [21] was used to scan SMART (SM00248) [22], ProSite (PS50088) [23], PRINTS (PR01415) [24] and PFAM (PF13606, PF00023) [25] for ankyrin repeat motif (ANK) annotations in all species. Further annotations were downloaded from the UniProt database [26]. The databases use slightly different algorithms resulting in variation in the number as well as the length and coordinates of annotations between them (Fig 3A). Accordingly, we retrieved all ankyrin repeat annotated sequences found in Swiss-Prot reviewed proteins from the following databases: UniProt (7,230 ankyrin repeats), SMART (6,396), ProSite (4,119), PRINTS (796) and PFAM (288) (Fig 3B) resulting in a total of 18,825 ANK annotations. After redundancy filtering, we established a high-quality set of 7,407 ankyrin repeat sequences: 4,109 (ProSite), 2,313 (SMART), 972 (UniProt) and 10 (PFAM) (Fig 3C) for analysis.

### Multiple sequence alignment

Several approaches were tried to align the 7,407 ankyrin repeat (AR) sequences, both sequence and structure-based. These included ClustalΩ [28], HMMER [29], T-Coffee [30], AMPS [31], Muscle [32] and STAMP [33]. When applied to all 7,407 sequences, these aligners introduced many gaps and a high proportion of misaligned residues which were inconsistent with known key residues in the ankyrin repeat. Accordingly, the final multiple sequence alignment (MSA) was obtained by carrying out a series of sequences-to-profile multiple sequence alignments with ClustalΩ (Fig 4) as follows.

First, the sequences were divided into different groups according to their length and database of origin. Then, sequences that had the most common length, 33 residues, coming from ProSite, were aligned using ClustalΩ version 1.2.2 with defaults. Sequences introducing gaps in the 33 high-occupancy columns were removed and re-aligned with a ClustalΩ sequence-to-profile alignment. Sequences inserting gaps yet again were removed from the alignment.

The remaining sequence groups, as defined by sequence length and database, were aligned to this growing alignment by consecutive sequence-to-profile alignments. As with the first

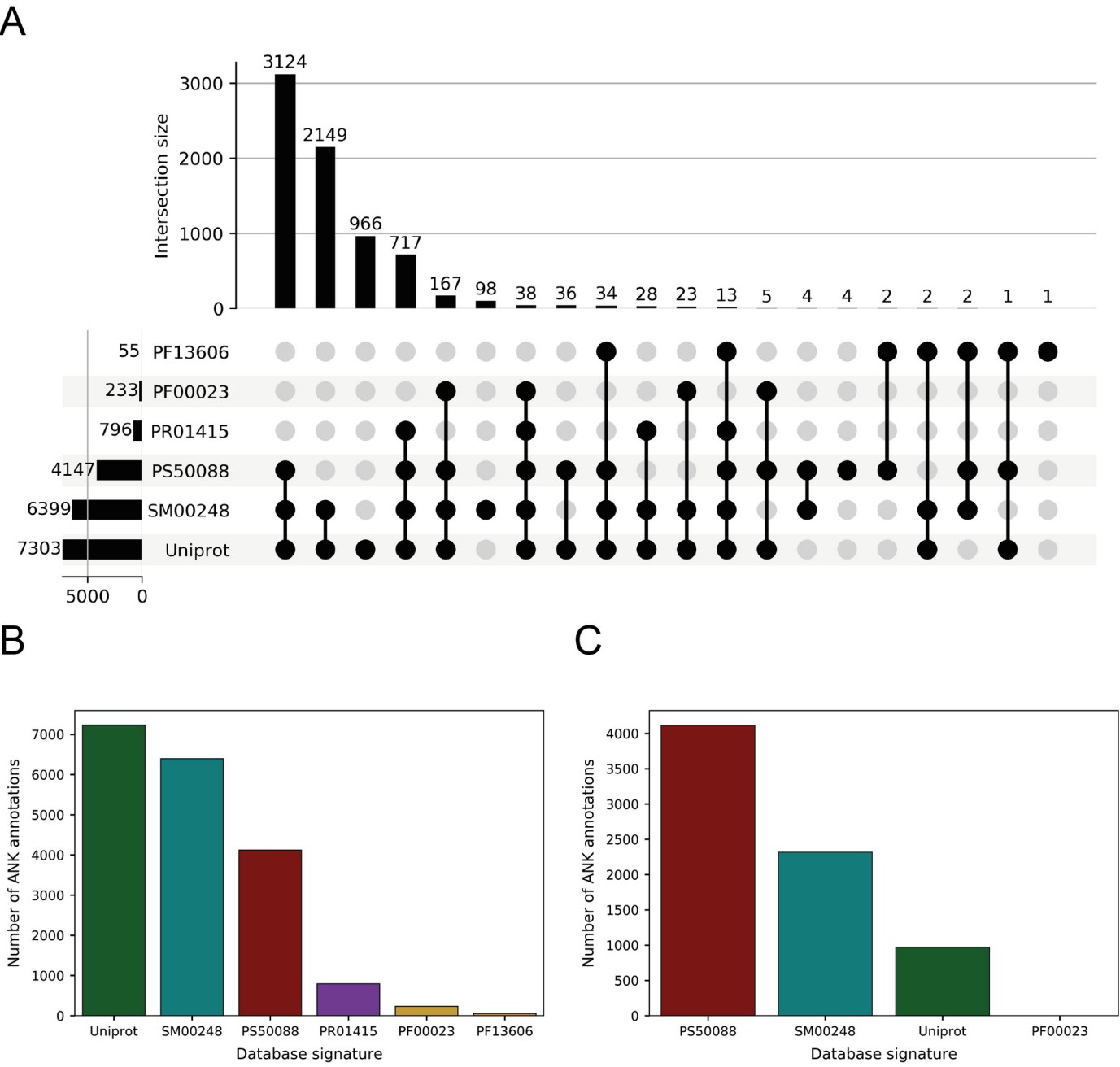

**Fig 3.** (A) Upset plot [27] showing the distribution of ANK annotations and the overlap between different database signatures. The vertical bar plot shows the total number of repeat annotations per database signature, whereas the horizontal one represents the number of annotations that are shared by the different signatures, i.e., the intersection between different sets of repeat annotations. For example, 3,124 out of the 7,303 repeats annotated by UniProt are shared with PS50088 and SM00248. Most of the annotations are shared between UniProt, SM00248 and PS50088. UniProt presents ≈ 1000 unique annotations which are not present in any other database; (B) This bar plot indicates the number of ANK annotations per database signature: 7,230, 6,396, 4,119, 796, 233 and 55 from left to right; (C) This bar plot shows the composition of the dataset resulting from the database merging, with ProSite accounting for ≈ 55% of the annotations, SMART for ≈ 30% and UniProt for the last ≈ 15%.

alignment, gap-introducing sequences were re-aligned and removed if necessary, for each group.

At the end of this process, ≈ 98% of the sequences were aligned in the resulting MSA. The remaining 2%, which comprised the gap-introducing sequences removed during the re-

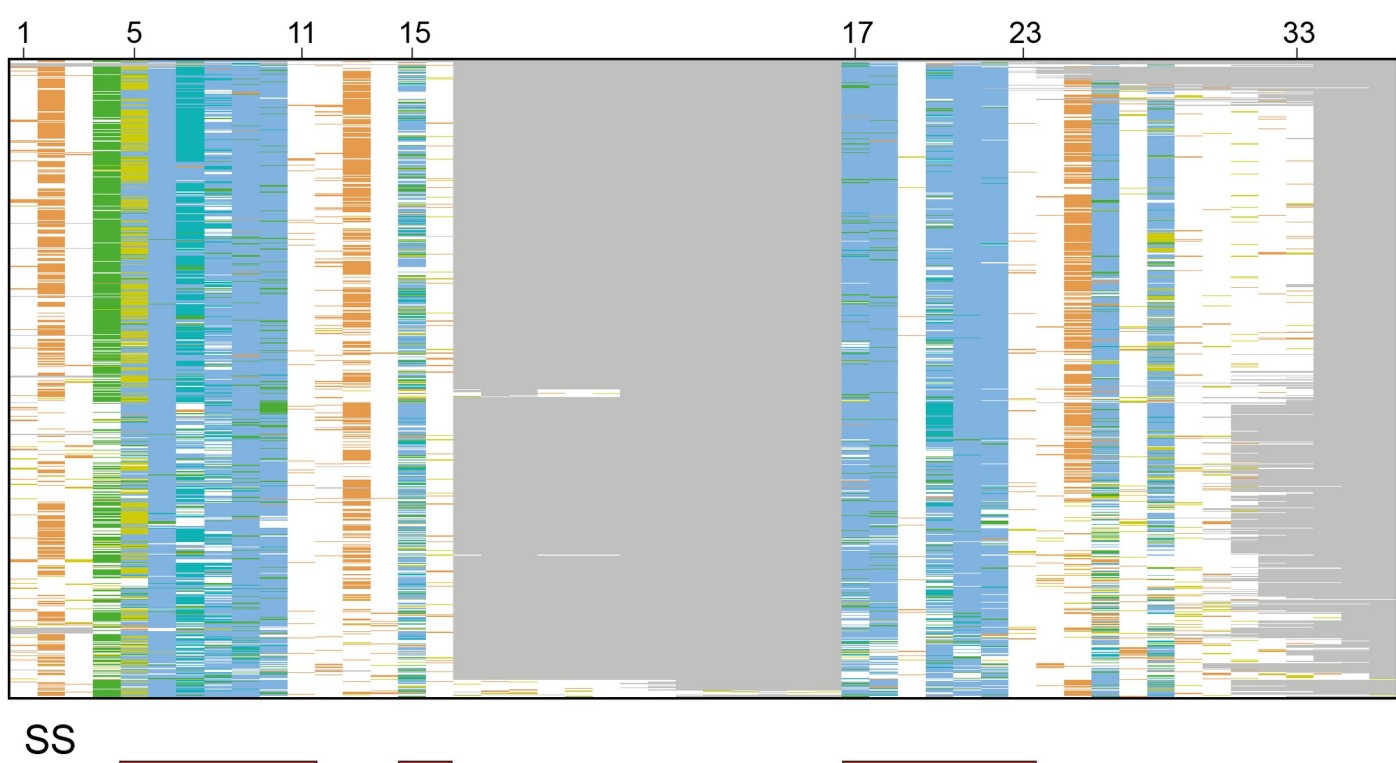

**Fig 4. Overview of the resulting MSA, including the 7,404 ankyrin repeat sequences.** Only columns with occupancy > 0.5% are shown. Sequences are sorted by a tree generated in Jalview using the average distance method and the BLOSUM62 matrix. Columns between 16–17 and after 33 represent insertions in some ankyrin repeats. Red boxes below the overview indicate the location of the secondary structure elements (SS), α-helices in this case, within the alignment. Grey dashed lines represent gaps and are mostly found at low-occupancy columns. Columns are coloured according to the ClustalX colour scheme [34]. Hydrophobic residues are coloured in blue, glycines in orange, prolines in yellow, polar residues in green and unconserved columns are coloured in white. Obtained with Jalview [35].

alignment phase of the process, were re-aligned to the main alignment by a profile-to-profile alignment shown as an overview in Fig 4.

## VarAlign and ProIntVar

A total of 35,691 variants found in the genome aggregation database (gnomAD) [20] coming from 1,435 human sequences were mapped to the MSA through VarAlign [18] (Fig 5). gnomAD contains exomes and genomes from a total of 141,456 unique unrelated individuals in control groups sequenced as part of various disease-specific and population genetic studies. Individuals affected by severe paediatric diseases were removed from the set as well as their relatives. For this reason, this dataset can be used as a general population control since disease variants might be present at an equivalent or lower frequency than in the general population.

419 sequences in the alignment were mapped to 209 different structures solved by X-ray crystallography in the PDB [36–38] via SIFTS [39] through VarAlign and ProIntVar [40]. These sequences correspond to 419 unique ankyrin repeats, found in 80 proteins. The real-space R value (RSR) and RSR-Z scores, as well as the real-space correlation coefficient (RSCC) quality metrics, as calculated by [41], were retrieved by ProteoFAV [18] from the validation reports in PDBe. Only residues with RSCC > 0.85 and RSRZ < 2 were considered for analysis. After this filtering step, our structural dataset comprised 383/419 unique ARs coming from 176/209 PDBs, representing 73/80 proteins. This dataset included 11,186 of the 13,059 residues with structural coverage before quality filtering. The average RSRZ per residue after filtering was −0.11 and the mean RSCC had a value of +0.95.

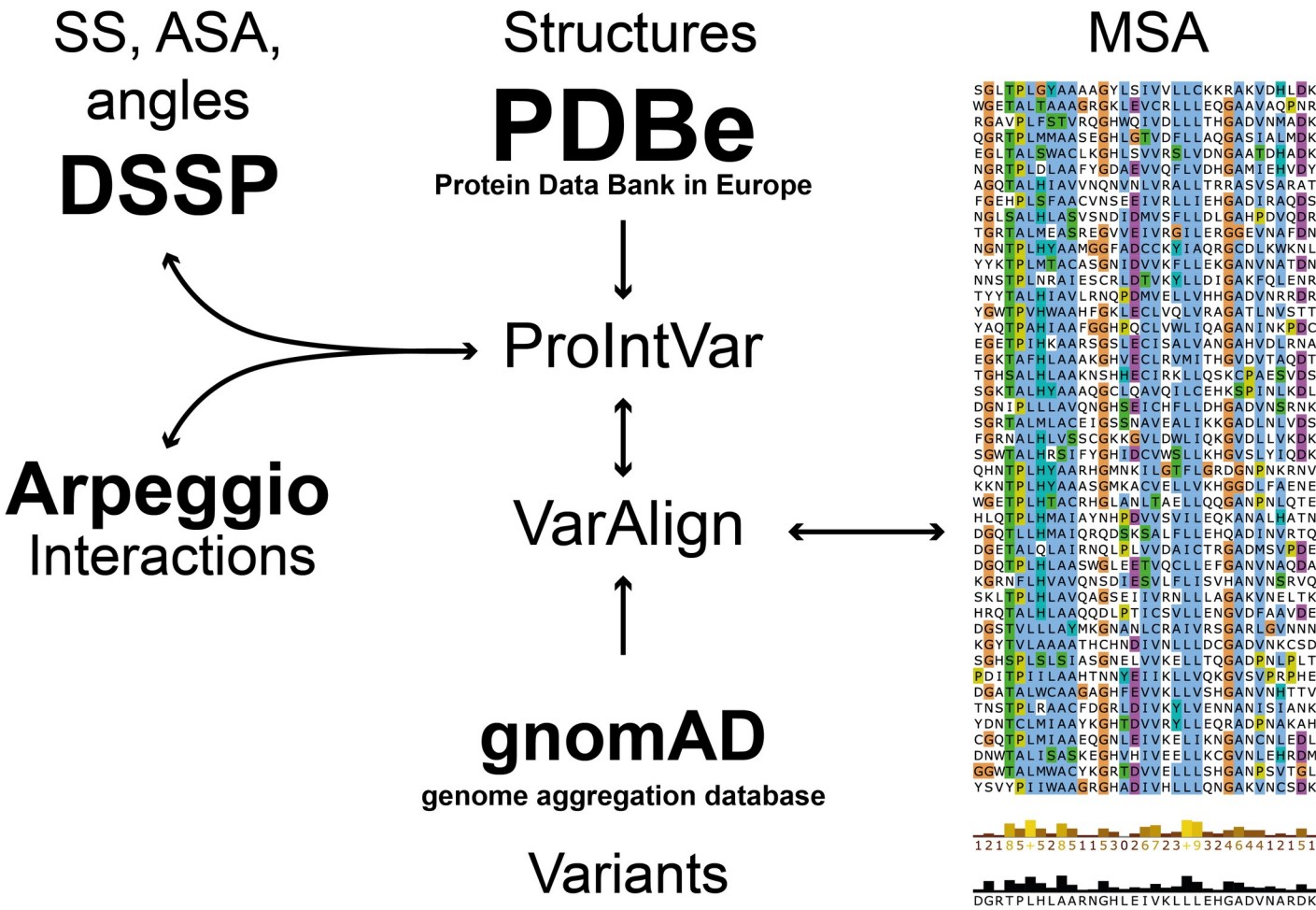

**Fig 5. Diagram showing the main components of the pipeline.** VarAlign retrieves variants found in human sequences in the MSA from gnomAD. ProIntVar retrieves structures from the PDBe and runs DSSP and Arpeggio to get secondary structure, accessible surface area and inter-atomic contacts information. Everything is mapped back to the residues and MSA columns [40].

DSSP [42] was run on all structures via ProIntVar and information from 381 ankyrin repeat sequences was used to determine the consensus secondary structure as well as the relative solvent accessibility (RSA) classification for all positions in the ANK, as described in MacGowan *et al* [18].

### Sequence divergence score

The Shenkin divergence score was used to characterise residue conservation at an alignment position [43]. This is a divergence score, based on Shannon's entropy (Eqs 1 and 2).

$$V_{Shenkin} = 2^S \times 6 \tag{1}$$

$$S = -\sum_{i}^{K} p_i \log_2 p_i \tag{2}$$

Where $S$ is Shannon's entropy and $i$ is every one of the $K = 20$ different amino acid types. The range of this diversity score is determined by Shannon's entropy. In a completely conserved alignment column, one amino acid residue will be found with a frequency of 1.0,

whereas the rest will not be present, resulting in an entropy of 0.0, and a minimum $V_{Shenkin} = 2^0 \times 6 = 6$. At the other extreme, an alignment column with all 20 amino acids at a frequency of 1/20 = 0.05 would give an entropy of $S \approx 4.32$, resulting in a maximum $V_{Shenkin} = 2^{4.32} \times 6 \approx 120$. Thus, low Shenkin scores indicate higher conservation at a position and *vice versa*. To simplify the interpretation of the score, we normalised the Shenkin score to 0–100 (Eq 3).

$$N_{Shenkin} = (V_{Shenkin} - V_{Shenkin_{min}})/(V_{Shenkin_{max}} - V_{Shenkin_{min}}) \tag{3}$$

Where $V_{Shenkin_{min}}$ is the score of the most conserved column within the alignment, Position 9 with a Shenkin score of 15.43 and $V_{Shenkin_{max}}$ is the score of the most diverse position, Position 3 with a score of 103.96.

## Enrichment in variants

The human genetic variants from gnomAD [20] were mapped to the MSA and missense variant enrichment scores (MES) were calculated for the 33 positions of the ANK. MES is expressed as the natural logarithm of an odds ratio (OR) and it represents the enrichment of variants in an alignment column relative to the average for the other columns. Columns were classified as depleted, enriched or neutral according to this MES [18]. 95% confidence intervals and p-values were calculated to assess the significance of these ratios [44].

As previously stated, gnomAD can be used as a general population control. It is expected that most of the variants observed in gnomAD do not have a detrimental effect on the fitness, i.e., are neither deleterious nor pathogenic. ClinVar [45] is a public archive of reports of the relationships among human variations and phenotypes. We retrieved all human missense variants mapping to the sequences in our alignment, but only found 22 variants classified as pathogenic. Variants seemed to be uniformly distributed within the ANK motif, however the uncertainty of these measures was too big due to the reduced size of the dataset. Consequently, we did not look any further into pathogenic variants.

## Enrichment in protein-substrate interactions

For the structural analysis, the biological units were retrieved from the PDBe. These are the preferred assemblies for each structure, instead of the asymmetric units, which might not reflect the packing of the protein observed in nature. They are computed based on the buried surface area and interaction energies as defined by PISA [46]. All inter-atomic contacts were calculated by Arpeggio [47]. Atoms were considered to interact if they were within 5Å of each other.

We considered all interactions between an ankyrin repeat and any different protein substrate present in the preferred assembly as protein-protein interactions (PPIs). A log enrichment score was calculated for PPIs per position in the motif in a similar manner to MES above. It is referred as protein-protein interaction enrichment score (PPIES). The number of protein-protein interactions per alignment column was normalised by the structural coverage of that column in structures presenting an interaction between an ARD and a bound peptide substrate. We considered that there was evidence of contact between an AR position and a bound peptide substrate if there was at least one inter-atomic contact involving the repeat position and the substrate in at least one of the structures representing the complex.

## Enrichment in intra-repeat contacts

A contact map, shown as a 33 × 33 matrix, for the 33 positions in the ANK, was calculated to show how often two positions interact within an AR. Each cell shows the proportion of repeats,

where evidence of contact between a given pair of residues has been observed. The absolute frequency is normalised by the coverage of a given pair of residues within a repeat. This intra-repeat contact map is symmetric.

Enrichment in intra-repeat contacts per position was calculated. Since the intra-repeat contact matrix is symmetric, the total number of contacts per residue, $C_i$, was calculated using Eq 4, where $c_{i,j}$ is the absolute frequency of contacts between any two amino acid residues present at positions $i$ and $j$ within the $K = 33$ positions in the ANK. The same approach was used to calculate the total structural coverage per ANK position, $O_i$, (Eq 5) where $o_{i,j}$ is the absolute frequency of both positions $i$ and $j$ being present in the same repeat.

$$C_i = \sum_{j=1}^{K} c_{i,j} \tag{4}$$

$$O_i = \sum_{j}^{K} o_{i,j} \tag{5}$$

The total number of contacts and coverage of the entire motif were calculated as the sum of all the contacts and coverages of the 33 positions, respectively (Eqs 6 and 7).

$$O_t = \sum_{i}^{K} O_i \tag{6}$$

$$C_t = \sum_{i}^{K} C_i \tag{7}$$

Enrichment in these contacts was calculated per position in the same fashion as for variants and PPIs. The same analysis was carried out on inter-repeat contacts. Nevertheless, no apparent relation between enrichment in inter-repeat contacts and conservation across homologues, nor depletion in missense variants was observed.

## Results and discussion

In this work, 7,407 ankyrin repeat sequences, including both human and other species, were used to build a multiple sequence alignment and conservation profile of the motif. Human genetic variation data coming from 1,435 human ankyrin repeats were used to study the distribution of variation within the motif. Moreover, 176 three-dimensional structures, representing a total of 383 different ankyrin repeats were used to structurally characterise in detail this motif by secondary structure, residue solvent accessibility, intra-domain contacts and protein-protein interactions. For the first time, human population variation data was used to explain in detail the evolutionary constraint acting upon this family of protein repeats, integrating these data with structure and sequence divergence.

### Conservation profile

The conservation profile derived from our MSA agrees with previous work [13]. Fig 6A shows the normalised Shenkin divergence score per position in the motif. As described in Methods, this score goes from 0–100. Among the most conserved positions ($N_{Shenkin} < 25$) we find Thr4, Pro5, Leu6, belonging in the TPLH motif, 4–7 as well as Ala9, Gly13, Leu21 and Leu22. Some of the most evolutionary diverse positions, on the other hand, include positions 1, 3, 11, 12 and 33 among others, all presenting $N_{Shenkin} > 75$. Most of the highly diverse positions are

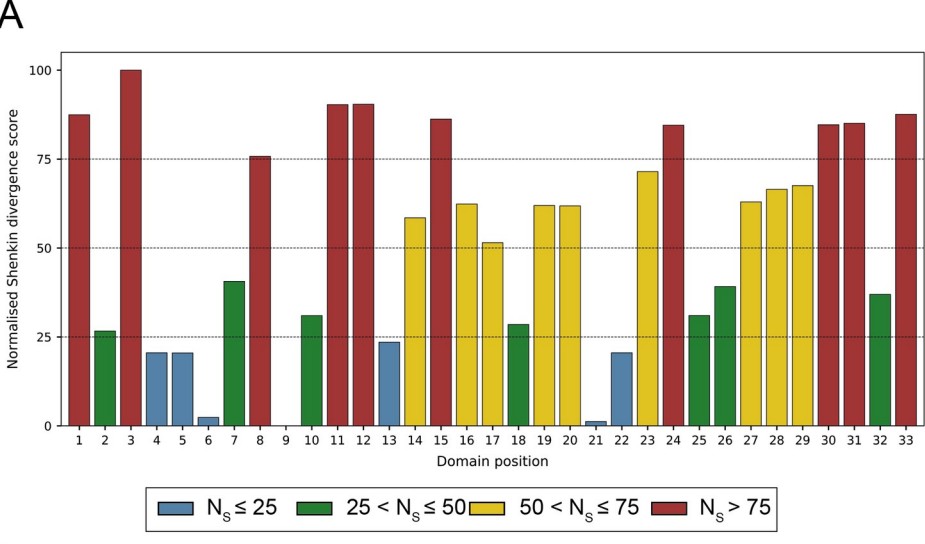

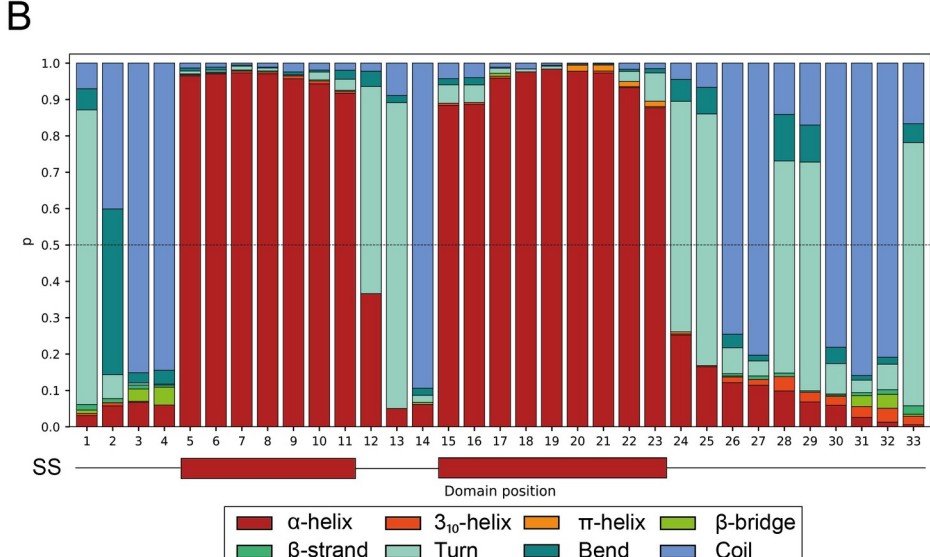

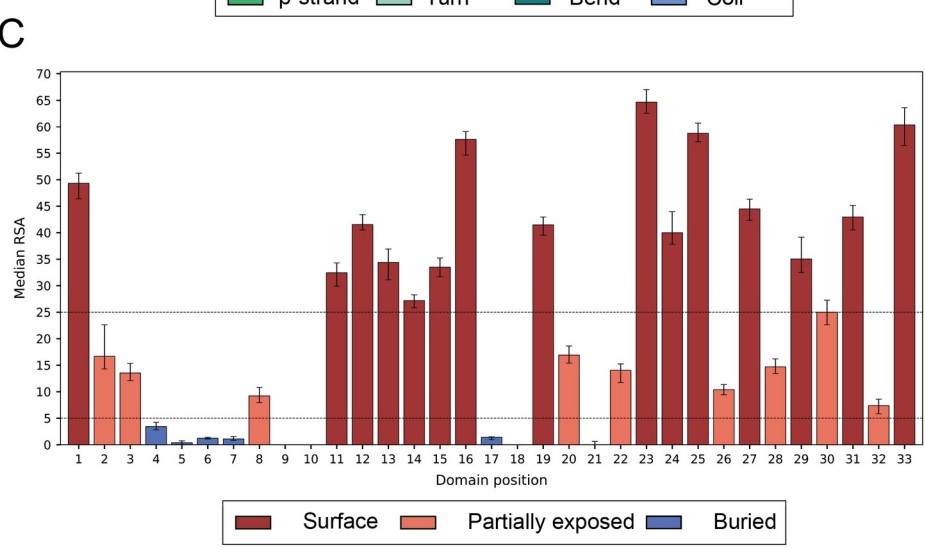

**Fig 6.** (A) Normalised Shenkin divergence score per domain position (Eq 3) calculated from the MSA containing 7,404 sequences. Positions are coloured according to their normalised Shenkin score as the legend indicates; (B) Secondary structure assignment per position. Within each position, each coloured bar represents the frequency of the eight states defined by DSSP: α-helix, $3_{10}$-helix, π-helix, β-bridge, β-strand, turn, bend and coil, observed for the residues with structural coverage at that column in the MSA. Most helices range from 5–11 and 15–23 and finish in 5-turns, usually at positions 12–13 and 23–24. Two β-turns are observed at positions 28–29 and 33–1; (C) Median residue relative surface accessibility per position, calculated from DSSP's accessible surface area [42] as described in Tien *et al* [48]. Error bars indicate 95% CI of the median. Positions were classified according to the specified thresholds: surface (RSA $\geq$ 25%), partially exposed (5% < RSA < 25%) or buried (RSA $\leq$ 5%) [49].

found on the concave surface and contribute to the variable interface where most of the substrate binding takes place.

## Secondary structure

Fig 6B shows the secondary structure assignment of the 33 positions in the ANK. Most of the repeats present a seven-residue long first helix ranging from the fifth to the 11[th] position and a second helix that in most cases is nine-residues long and extends from the 15[th] to the 23[rd] position. Our results also show four turns along the ANK. Two of these turns, found at positions 12–13 and 24–25, are 5-turns and simply indicate the end of the α-helices, whereas the other two, positions 28–29 and 33–1, are β-turns. These two β-turns were classified as type I β-turns according to the φ and ψ dihedral angles distribution of consensus columns 33, 1 and 28, 29 [50]. Positions 27 and 32 in the alignment present either Asn or Asp with a high frequency of 44% and 58% respectively. Consequently, we classified the turns they initiate as Asx motifs (Fig 7). The turn at positions 27–30 was classified as an Asx-β-turn and the one at 32–2 as a type 1 β-bulge loop with an Asx motif [51]. Repeats that do not have Asx at positions 27 and 32, form a simple β-turn, instead of an Asx motif since they lack the extra hydrogen bonds that this secondary structure motif requires. The conservation of these Asx residues on both turns, suggests a structural relevance and role of these Asx motifs on the correct packing of the ARD.

## Relative solvent accessibility and surface classification

The surface of the ankyrin repeat domain has previously been divided into two faces: concave (positions 32–12) and convex (positions 13–31) (Fig 8A and 8B) [52]. Positions with high RSA (RSA ≈ 50%), such as 1, 12 and 33 are found near positions 13 and 32. Due to their high solvent accessibility, these positions were used to define ridges at the limits of the concave and convex surface. However, our analysis of all available structures also showed positions 23 and 25 to have a high RSA (Fig 6C). In addition, in the same fashion as positions 1, 33 and 12, positions 23 and 25 from different repeats form a ridge on the domain structure. This ridge suggests the definition of a third surface of the domain or basal surface as shown in Fig 8C and enabled the classification of all positions that were not buried into one of the three defined surfaces, (Table 1). This classification is shown in Fig 8C for an ARD containing 12 repeats [53].

Some blue-coloured regions, representing buried residues, can be observed on the ARD surface on Fig 8C. These are the side chains of buried residues within the motif dominated by Thr4 and His7. The correct classification of the positions in the ANK as either buried or any of the defined surfaces is critical to calculate accurate enrichment scores in missense variants and protein-protein interactions on a surface basis later in the analysis.

## Missense variants enrichment analysis

21,338 missense variants from 1,435 human ankyrin repeat sequences were used to calculate column-specific missense enrichment scores (MES). The MES measures how enriched in

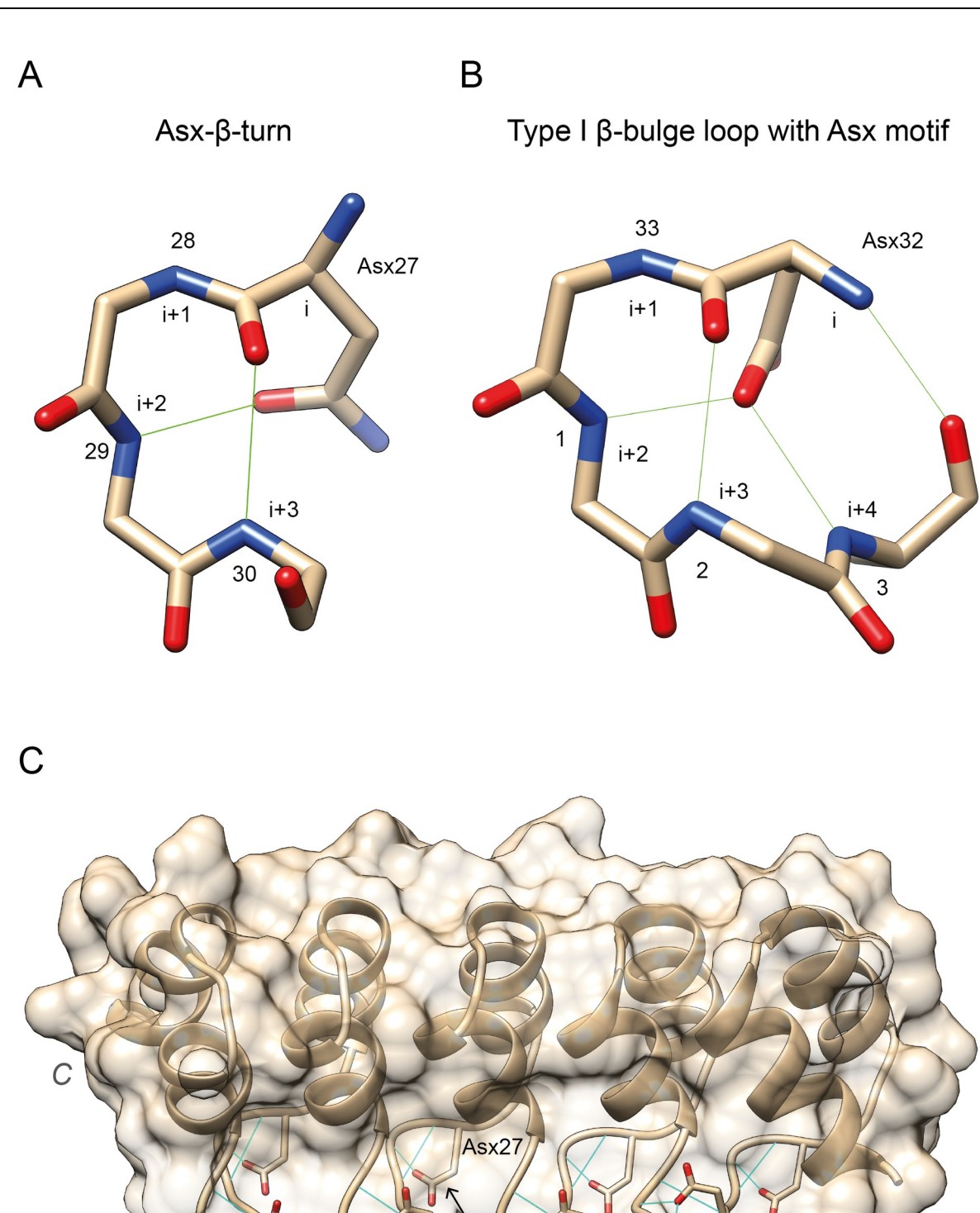

**Fig 7. Hydrogen bonding patterns of the two Asx motifs found in the ANK and their location within the ARD.** Only repeats with either Asn or Asp at these positions will present this hydrogen bonding pattern. (A) Asx-β-turn at positions 27–30. Conserved Asx, i.e., Asp/Asn, side chain at position i = 27 forms an extra hydrogen bond with backbone N at position i + 2; (B) Type 1 β-bulge loop with Asx motif at positions 32–3. Conserved Asx side chain at domain position i = 32 forms two hydrogen bonds with backbone N of residues i + 2 and i + 4. The rest of the hydrogen bonds originate from the backbone of the residues and are not specific of Asx motifs. PDB: 5MA3 [14]; (C) DARPin-8.4 (Barandun J, Schroeder T, Mittl PRE, Grutter MG) PDB: 2Y1L. Light blue lines represent the hydrogen bonds that determine these secondary structure motifs. The conservation of the Asx residues at positions 27 and 32, and the hydrogen bonding network they facilitate, suggest that these Asx motifs are one of the most structurally important components of the ankyrin repeat domain structure. Figure obtained with UCSF Chimera [7].

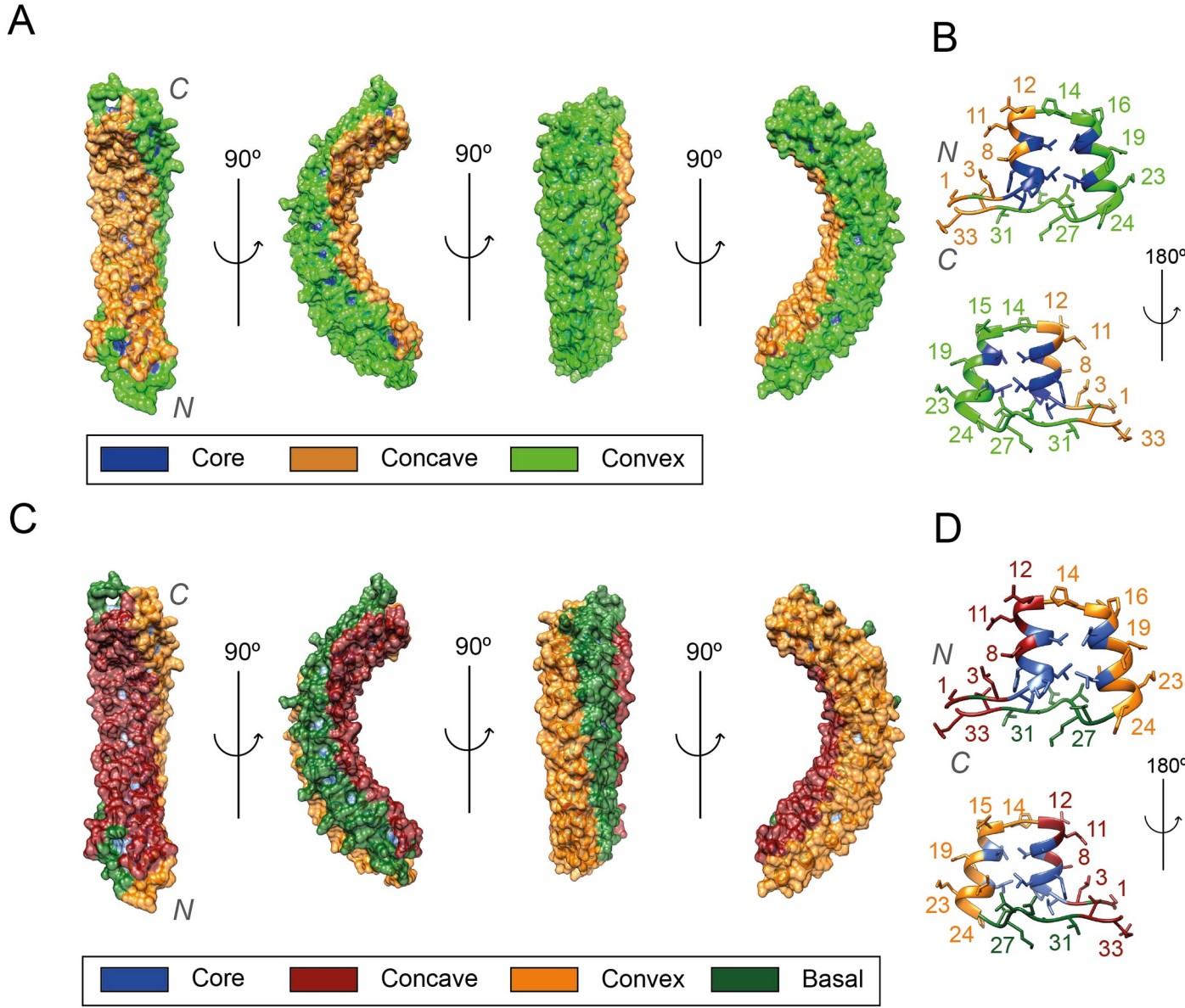

**Fig 8.** Comparison of the original definition of the ARD surfaces (A, B) with the new definitions derived from the results of this study (C, D). All panels refer to the D34 region of ANK1 ARD, PDB accession: 1N11 [53]. This structure shows 12 out of the 23 ARs found on this ARD; (A) Surface of an ARD. Residues conforming the concave surface are coloured in orange, residues on the convex surface in green and buried residues in blue; (B) Structure of an individual repeat. The first α-helix and the β-turn region form the concave surface, whereas the second helix and the loop form the convex one; (C) Residues forming the concave surface are coloured in dark red, residues on the convex surface in orange, the basal surface is coloured on dark green and buried residues in blue; (D) Structure of an individual repeat with new surface classification. Figure obtained with UCSF Chimera [7].

**Table 1. Classification of the 33 positions within the ANK in the different surfaces.**

| Surface | Consensus residue positions |
|---|---|
| Core | 4, 5, 6, 7, 9, 10, 17, 18, 21 |
| Concave | 1, 2, 3, 8, 11, 12, 32, 33 |
| Convex | 13, 14, 15, 16, 19, 20, 22, 23, 24 |
| Basal | 25, 26, 27, 28, 29, 30, 31 |

missense variants an alignment column is compared to the average of the other columns in the alignment [18]. The 33 columns of the motif were classified into four categories according to their normalised Shenkin divergence score ($N_{Shenkin}$) and MES. Columns with $0 \leq N_{Shenkin} \leq 25$ and MES < 0 were classified as conserved and missense depleted (CMD), whereas columns satisfying $0 \leq N_{Shenkin} \leq 25$ and MES > 0 were called conserved and missense enriched (CME). We also classified those columns with $75 \leq N_{Shenkin} \leq 100$ as unconserved and either missense depleted (UMD) if MES < 0 or enriched if MES > 0 (UME). $N_{Shenkin}$ ranges from 0 for the most conserved (Position 9) to 100 for the most divergent (Position 3) column within the ANK alignment. Fig 9A shows the enrichment in human population missense variants per position in the ANK relative to their Shenkin divergence score. Positions that are depleted in missense variants relative to the rest of positions within the ANK are the most interesting and are likely to be functionally important. Depletion in missense variation represents population constraint within the human population and is therefore indicative of functional and/or structural relevance [18].

The relationship between residue solvent accessibility and enrichment in missense variants was examined. As expected, on average, buried residues (RSA ≤ 5%) were depleted in missense variants relative to residues present on the surface ($MES = -0.10$, $p = 1.9 \times 10^{-7}$). Furthermore, residues present on the concave surface of the ankyrin repeat domain were significantly depleted in missense variants relative to the other surfaces, ($MES = -0.08$, $p = 4.4 \times 10^{-4}$). The convex surface was neither enriched nor depleted, whereas the basal surface was significantly enriched in missense variants: ($MES = 0.09$, $p = 6.2 \times 10^{-6}$). Moreover, the basal surface is significantly enriched in missense variation relative to the convex one ($MES = 0.08$, $p = 8.8 \times 10^{-4}$). These results can be observed in structure in Fig 9B and are further discussed in *"Different surfaces of the ARD"* below.

## Ankyrin repeat contact maps and enrichment

In this work, contacts across all known ankyrin repeat structures were considered instead of just a single repeat or domain. This allowed a comprehensive contact map for the repeat motif to be calculated as well as enrichment scores for each residue's contacts within the repeat, thus highlighting the most structurally important positions.

## Intra-repeat contacts

Fig 10A shows the symmetric contact matrix that defines the ankyrin repeat motif. Contacts between residues within 2–5 amino acids of each other are around the diagonal. Most other contacts are between residues along the first and the second α-helices, from positions 5–11 and 15–23, respectively or contacts between residues close in sequence within the loops. This pattern of contacts is typical of helices or turns. We focused on contacts most relevant to the ANK fold, i.e., helix-helix contacts, by filtering out contacts between positions within ≤ 6 residues of each other. Fig 10B shows the enrichment in these intra-repeat contacts for each position within the ANK. CMD positions are among the most enriched in intra-repeat

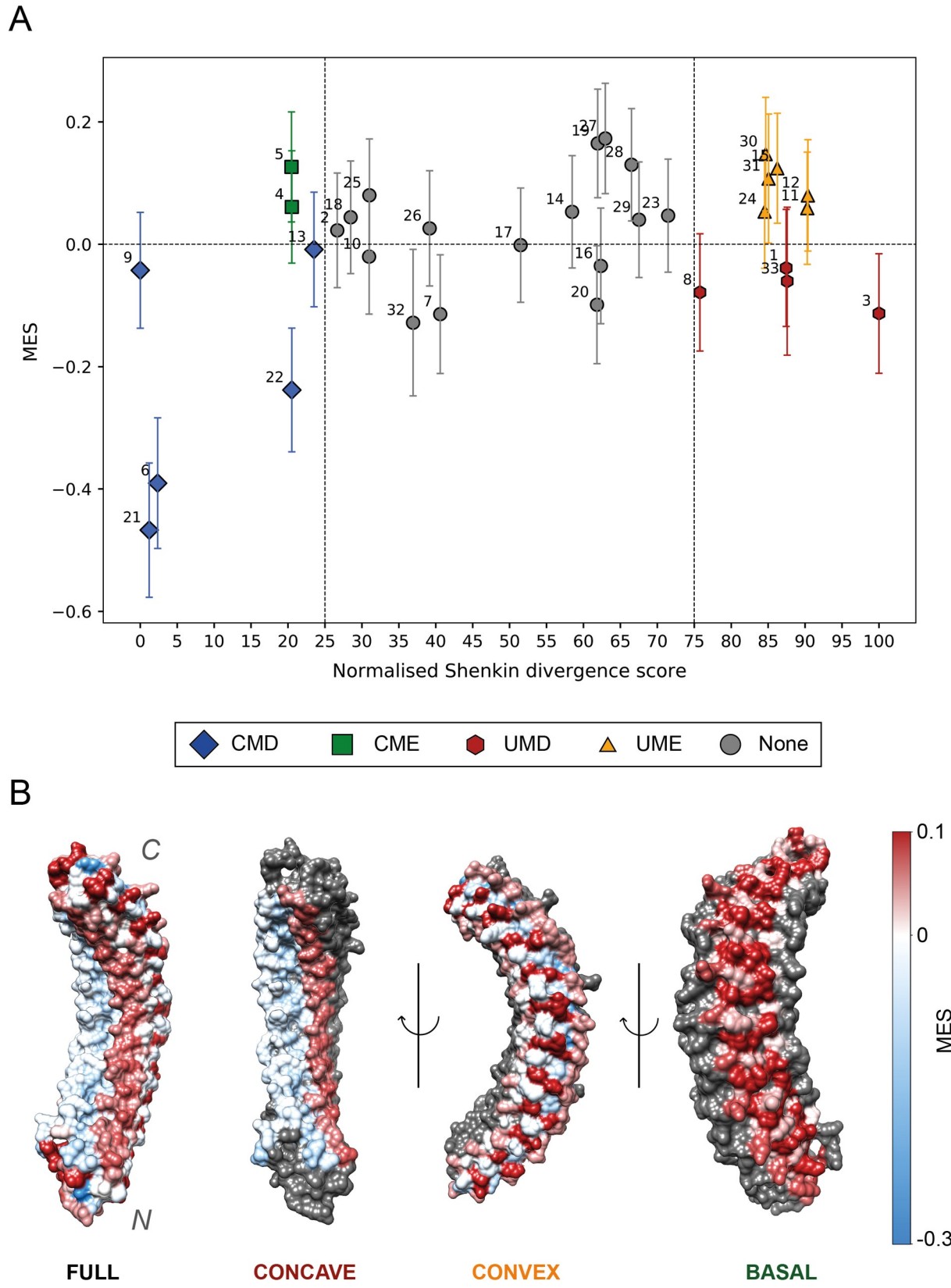

**Fig 9.** (A) Relative Missense Enrichment Score (MES) against normalised Shenkin divergence score for the 33 positions of the domain. Blue diamonds: CMD positions (6, 9, 13, 21, 22); Green squares: CME positions (4, 5), UMDs are coloured in red hexagons (1, 3, 8, 33) and UMEs in orange triangles (11, 12, 15, 23, 24, 30, 31). Error bars represent 95% CI of the MES, i.e., ln (OR). Positions coloured in grey circles are classed as "None", for they do not meet our divergence score thresholds; (B) D34 region of ANK1 ARD, PDB accession: 1N11 [53] This structure shows 12 out of the 23 ARs found on this ARD. Residues are coloured according to the missense enrichment score of the alignment column they align to in the MSA. The colour scale goes from blue (missense-depleted) to red (missense-enriched) going through white (neutral). From left to right, the full domain, then concave, convex and basal surface are coloured. On each of the last three representations, only one surface is coloured. Residues that are not constitutive of the displayed surface are coloured in grey. Overall, the concave surface is coloured in a light blue colour (except positions 11 and 12), indicating its depletion in missense variants, relative to the other positions within the ANK. Figure obtained with UCSF Chimera [7].

interactions which suggests an important role in ankyrin repeat packing and may explain their conservation across homologues and depletion in missense variants within the human population.

## Protein-substrate interaction enrichment

Fig 11A shows the enrichment in Protein-Protein interactions (PPIs) per position in the ANK. Out of the 176 protein structures that satisfied our quality thresholds, as described in Methods, 63 include protein substrates. These represent the interaction between 35 different ARDs and their substrates, accounting for a total of 142 repeats. All the positions that are found on the concave surface are enriched in PPIs. This includes the highly conserved His7, which despite its overall burial, is partly accessible to the concave surface. Positions 13 and 14, which define the beginning of the convex surface, are also enriched in PPIs. This might be explained by their close proximity to the concave surface.

We also compared the enrichment in PPIs between different surfaces. As expected, buried residues were significantly depleted on average relative to surface residues, ($PPIES = -1.02$, $p < 10^{-16}$). Compared to residues belonging to other surfaces, concave residues are highly enriched in PPIs: ($PPIES = 1.86$, $p < 10^{-16}$). Conversely, convex and basal residues are both depleted in PPIs relative to residues present in the other surfaces: ($PPIES = -0.79$, $p < 10^{-16}$) and ($PPIES = -2.19$, $p < 10^{-16}$), respectively. In addition, the direct comparison between basal and convex showed that residues on the convex surface were enriched in PPIs relative those in the basal surface: ($PPIES = 1.31$, $p = 5.40 \times 10^{-15}$). All these differences in enrichment in PPIs between different surfaces can be observed in Fig 11B, which shows the different surfaces of an ARD, where residues are coloured according to the PPIES of the column they align to. These results agree with [8] and show the importance of the concave surface in substrate binding. They also illustrate the rare, though existing, convex binding as well as the practically null contribution of the basal surface to substrate binding.

## Different surfaces of the ARD

In this work, we define the binding mode of an ARD as given by the number of repeats and residues that bind the substrate, as well as the surface the latter belong to. These modes can either be absolute or combined/mixed. In the former, one surface dominates the binding, whereas in the latter, a combination of different surfaces accounts for most of the substrate binding residues. For this part of the analysis, only those proteins with a minimum of two repeats and four residues binding the substrate were considered. Of the remaining 25 proteins, 21 (84%) presented a concave binding mode. Only one protein (4%), Ankyrin repeat domain-containing protein 27, ANKRD27 (Q96NW4), presented a dominant convex binding mode (PDB: 4CYM) [54] and none presented a basal mode. The other three (12%) proteins presented a mixed binding mode, where concave, convex, basal and even buried residues participate in the substrate binding. Two of these proteins are EMB506 (Q9SQK3) and AKRP

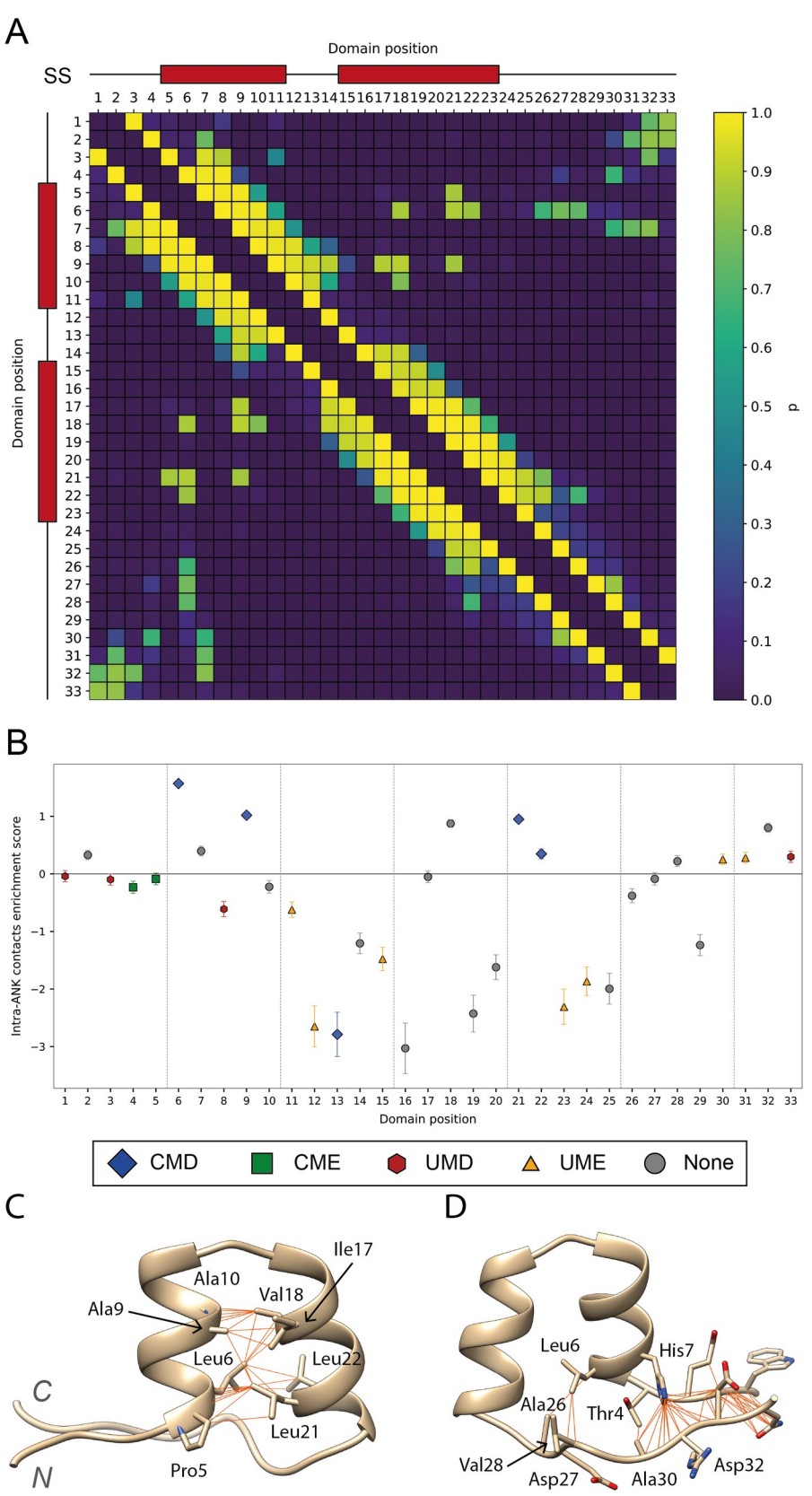

**Fig 10.** (A) Contact map for intra-repeat residue-residue interactions. Cells are coloured according to the probability of observing contact between two positions with the viridis colour palette. Red boxes above axis indicate the location of the secondary structure (SS) elements, α-helices, in the motif; (B) Intra-repeat contacts enrichment plot. Error bars indicate 95% CI of the enrichment score, i.e., ln (OR). Data points are coloured according to their missense enrichment and residue conservation classification (Fig 9); (C) Cluster of intra-repeat contacts between the first and second helices. Residues 5, 6, 9 and 10 in the first helix interact with residues 17, 18, 21 and 22 by forming hydrophobic interactions. These positions are all buried and conserved; (D) Cluster of intra-repeat contacts between the start and end residues of an AR. These interactions are not as specific as the ones in the first cluster and they include diverse positions such as 1, 3, 31 or 33. These are the most frequently observed contacts across all structure displayed in an example repeat. PDB: 5MA3 [14]. Figure obtained with UCSF Chimera [7].

(Q05753) and are found in *Arabidopsis thaliana*. The 3D structure PDB: 6JD6 depicts these two ARDs binding each other, unlike any other example in our dataset. In this complex, the protein substrate is another ankyrin repeat domain, and residues found in all surfaces participate in binding. The last example is Q7Z6K4 (NRARP), which interacts with the ARD of NOTCH1 (P46531) to form an even longer solenoid domain to bind their substrates in PDB: 6PY8 [55].

These binding surfaces present different patterns of enrichment in variation as well as protein-substrate interactions. The concave surface is significantly depleted in missense variants and enriched in PPIs, whereas the basal is the complete opposite and is enriched in missense variants and depleted in PPIs. These results confirm the dominance of the concave binding mode. In addition, we have observed that ARDs can also present a convex binding mode [54], whereas no basal binding mode was observed in the dataset. The differential importance in substrate binding seems to influence the distribution of missense variants within the motif.

## Conserved and missense depleted positions

Positions 6, 9, 21 and 22 were found to be highly conserved and depleted in missense variants relative to the other motif positions (CMD). These positions are mostly buried, and present hydrophobic residues. CMDs are enriched in intra-repeat contacts. This population-level constraint agrees with the amino acid conservation and supports the structural relevance of these residues. Positions 7 and 32 are not as conserved as the residues we have classified as CMD; however, they are significantly depleted in missense variants as well. These two residues are structurally important due to the hydrogen bonding networks they create, as can be seen in Fig 2C and 2E.

## Unconserved and missense depleted positions

It is known that the concave surface possesses high sequence variability, in order to accommodate the diversity of protein substrates that ankyrins bind [8]. Positions 1, 3, 8 and 33 are amongst the most diverse positions within the ANK, though at the same time depleted in missense variants in the human population. These positions are enriched in PPIs (*PPIES* = 3.6, $p < 10^{-16}$) and constitute most of the concave surface of the ARD. Missense depletion at these sites show that they are constrained at a population level, thus confirming the functional importance of these residues.

Fig 12 illustrates how the ARDs of the closely related pairs ANKRA2/RFXANK and TNKS1/TNKS2 bind their protein substrates. ANKRA2 and RFXANK are human proteins that are involved in the regulation of transcription by RNA polymerase II. Both ARDs contain five repeat units. The domains are very similar in sequence, including UMD and unconserved positions 11 and 12, which do not vary across these proteins' homologues. Multiple structures have been solved portraying the interaction between these ARDs and more than five different protein substrates. All the substrates present the shared binding motif PXLPX[I/L] [56,57]. A

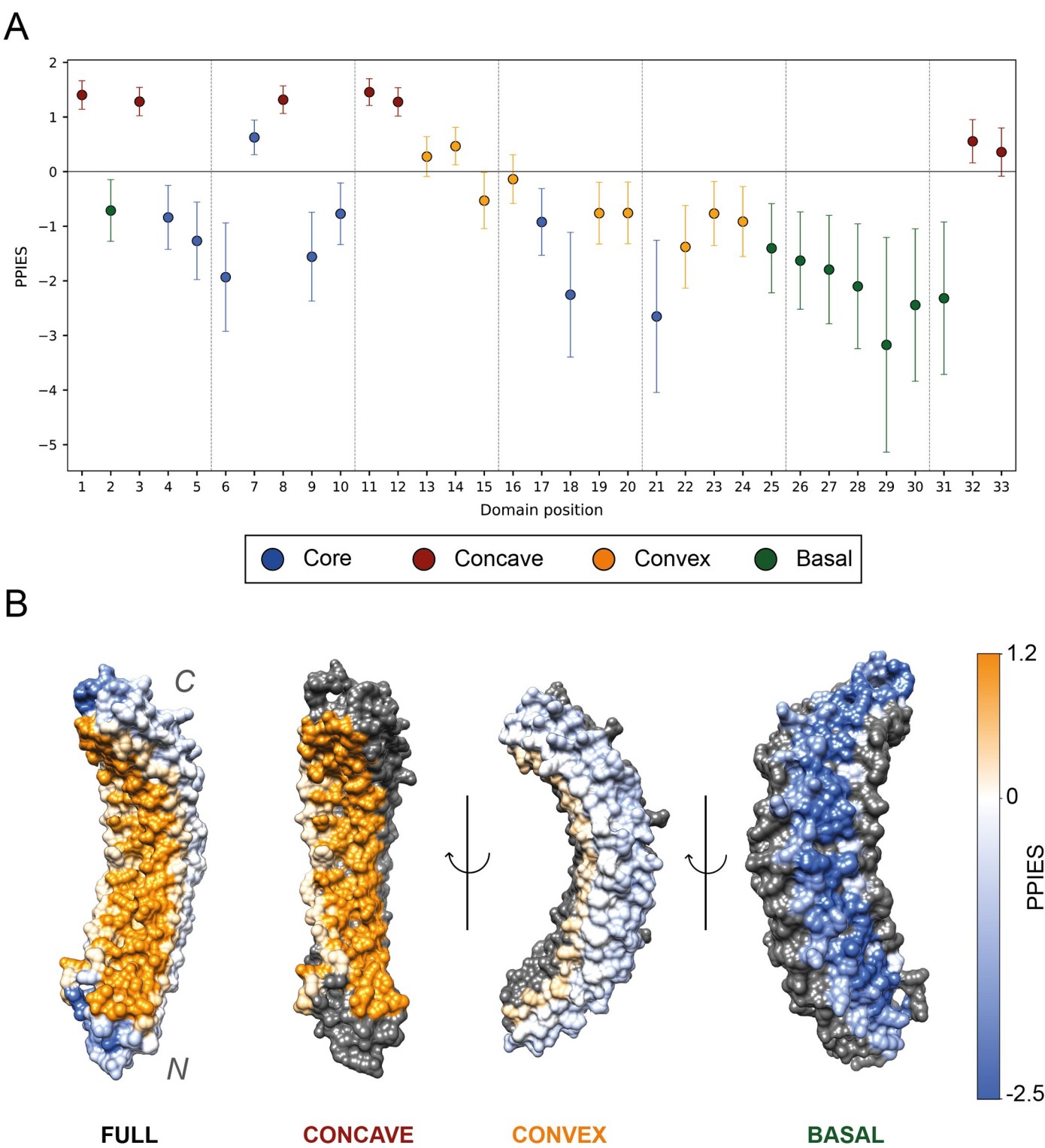

**Fig 11.** (A) Protein-substrate interactions enrichment plot. Error bars indicate 95% CI of the protein-protein interactions enrichment score (PPIES), i.e., ln (OR). Data points are coloured according to their surface classification (Table 1); (B) D34 region of ANK1 ARD, PDB accession: 1N11 [53]. This structure shows 12 out of the 23 ARs found on this ARD. Residues are coloured according to the PPIES of the alignment column they align to in the MSA. The colour scale goes from blue (depleted in PPIs) to orange (enriched in PPIs) going through white (neutral). From left to right, the whole domain, then the concave, convex and basal surface are coloured. On each of the last three representations, only one surface is coloured. Residues not belonging in that surface are coloured in grey. Overall, the concave surface is coloured in a strong orange colour, indicating its importance in protein binding, whereas the basal one presents a dark blue colour, indicative of its overall depletion in PPIs. Figure obtained with UCSF Chimera [7].

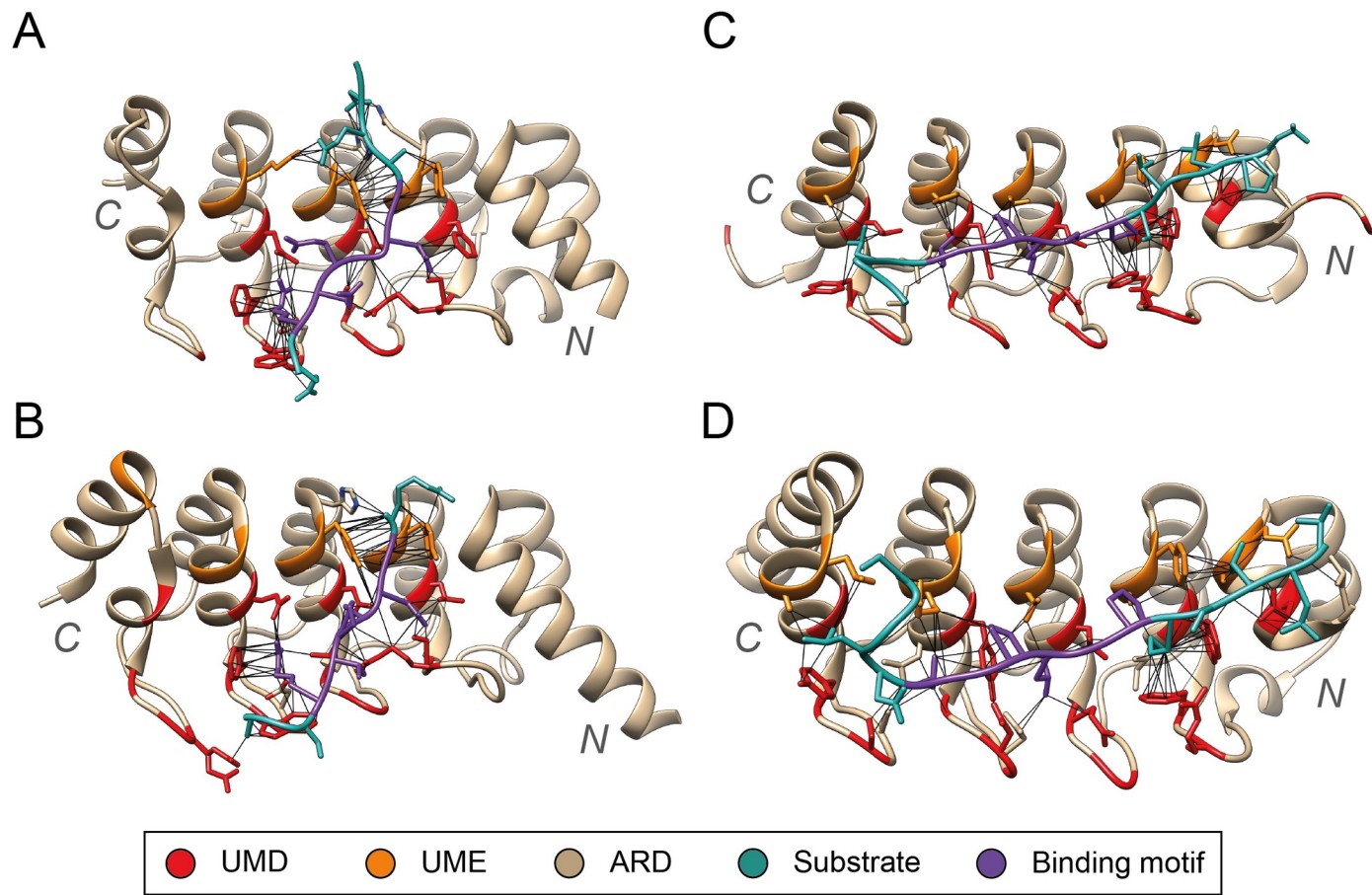

**Fig 12. ARDs in complex with substrates.** (A) RFXANK and RFX5 (PDB ID: 3V30) [56]; (B) ANRA2 and HDAC4 (3V31); (C) TNKS2 and ARPIN (4Z68) [59]; (D) TNKS1 and USP25 (5GP7) [60]. UMD positions (red) and UMEs 11, 12 (orange) are conserved across proteins that bind similar substrates (dark cyan). For example, these positions are conserved across TNKS2 and TNKS1, which are known to bind substrates with the motif RXXPDG (purple). Similarly, RFXANK and ANRA2, bind substrates with the motif PXLPX[I/L] (purple). Figure obtained with UCSF Chimera [7].

similar pattern can be observed with TNKS1 and TNKS2 and the substrates they bind, which share the tankyrase binding motif RXXPDG [58].

These examples show how ankyrin domains that present similar concave surfaces, determined by their UMD positions (1, 3, 8 and 33) bind similar protein substrates, or at least, substrates that share a binding motif. At the same time, it seems that all substrates binding these domains share a binding motif. These findings further support the hypothesis presented by MacGowan *et al* [18], which states that UMDs are determinant for substrate binding specificity.

## Conclusions

The multiple sequence alignment of homologues and the aggregation of genetic variants, or other features, over alignment columns, as described in MacGowan *et al* [18], can provide insight at the residue level on the evolutionary constraint acting on functional domains as well as highlight structural or functionally relevant residues in protein domains. Overall, a clear variation distribution pattern can be observed within the ankyrin repeat motif. There are five positions that are conserved and depleted in missense variation due to their structural

importance, e.g., enrichment in intra-repeat contacts. Four other positions are highly variable within the family and overall depleted in missense variants and are key for specific substrate binding.

In this study, we used 7,407 ankyrin repeat sequences, 21,338 human missense variants and 176 3D structures to study the distribution of missense variants within the ankyrin repeat motif and explain the observed patterns with structural data. The general conclusions are as follows.

1. Two of the turns found on the secondary structure of the ANK, positions 28–29 and 33–1, are Asx motifs. Positions 27 and 32 present conserved Asx.

2. The surface of the ARD can be divided in three different surfaces using the RSA of the repeat positions.

3. Positions that are conserved and depleted in missense variants (CMD) are significantly enriched in intra-repeat contacts ($OR = 2.8$, $p \approx 0$) and are key for the correct packing of the motif as well as the domain.

4. Positions that are unconserved yet depleted in missense variants (UMD) are heavily enriched in protein-protein interactions ($OR = 3.6$, $p < 10^{-16}$) and might be responsible for substrate binding specificity in the motif.

5. The concave surface of the ARD is significantly enriched in PPIs ($PPIES = 1.86$, $p < 10^{-16}$) and consequently depleted in missense variation ($MES = -0.08$, $p = 4.4 \times 10^{-4}$) whereas the other two surfaces are less constrained in line with their reduced importance in substrate binding.

## Acknowledgments

We thank Drs Marek Gierlinski, Matthew Parker and Jim Procter for their insightful suggestions during this research. We also thank Prof. Ulrich Zachariae for critical reading of the manuscript and the IT service of the University of Dundee for their support of our HPC infrastructure.

## Author Contributions

**Conceptualization:** Javier S. Utgés, Maxim I. Tsenkov, Noah J. M. Dietrich, Stuart A. MacGowan, Geoffrey J. Barton.

**Formal analysis:** Javier S. Utgés, Maxim I. Tsenkov, Noah J. M. Dietrich, Stuart A. MacGowan, Geoffrey J. Barton.

**Funding acquisition:** Geoffrey J. Barton.

**Investigation:** Javier S. Utgés, Maxim I. Tsenkov, Noah J. M. Dietrich, Stuart A. MacGowan, Geoffrey J. Barton.

**Methodology:** Javier S. Utgés, Maxim I. Tsenkov, Noah J. M. Dietrich, Stuart A. MacGowan, Geoffrey J. Barton.

**Project administration:** Geoffrey J. Barton.

**Resources:** Geoffrey J. Barton.

**Software:** Javier S. Utgés, Maxim I. Tsenkov, Noah J. M. Dietrich, Stuart A. MacGowan.

**Supervision:** Maxim I. Tsenkov, Stuart A. MacGowan, Geoffrey J. Barton.

**Visualization:** Javier S. Utgés.

**Writing – original draft:** Javier S. Utgés.

**Writing – review & editing:** Javier S. Utgés, Maxim I. Tsenkov, Stuart A. MacGowan, Geoffrey J. Barton.

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
