## [Decision Letter · Decision Letter 0]

18 Jul 2021

Dear Prof Barton,

Thank you very much for submitting your manuscript "Ankyrin Repeats in Context with Human Population Variation" for consideration at PLOS Computational Biology. As with all papers reviewed by the journal, your manuscript was reviewed by members of the editorial board and by several independent reviewers. The reviewers appreciated the attention to an important topic. Based on the reviews, we are likely to accept this manuscript for publication, providing that you modify the manuscript according to the review recommendations.

Sincerely,

Roland L. Dunbrack Jr., Ph.D.

Associate Editor

PLOS Computational Biology

Arne Elofsson

Deputy Editor

PLOS Computational Biology

[LINK]

Reviewer's Responses to Questions

**Comments to the Authors:**

Reviewer #1: The paper entitled “Ankyrin Repeats in Context with Human Population Variation” describes single nucleotide variation from a human population within a large ANK repeat superfamily. They interpret the variation in the context of ANK conservation and structure and suggest their methods would apply to understanding other protein families. The authors present a great deal of introductory material on known characteristics of the repeat and their data generally support the known characteristics of the fold. The novelty includes introduction of a basal surface based on RSA as well as interpreting positions enriched and depleted in missense mutations. Overall, the work is presented clearly with a few minor points elaborated below.

I think the strategy of combining all repeats into one MSA is useful in that it “boosts the statistical power” as the authors point out. Unfortunately, the use of repeats makes the method less appealing for all folds in general, and the author statement should be clarified accordingly. It is also interesting in that the use of repeat MSA probably removes signals from ortholog conservations, allowing insight into specificity. Perhaps more attention should be paid to the relative content of the MSA in terms of ortholog vs. paralog, vs repeat. Finally, the examples are informative and represent the most common binding surface. However, examples of binding to the other surfaces might also be informative, especially for the newly introduced basal surface. Perhaps it may add non-specific interaction surface that is robust to mutation? Perhaps it allows for adaptive evolution?

I think the materials used for the analysis should be provided, including the MSA as well as lists of the PDBs.

Minor comments:

line 69 and &1 are missing commas, please check the manuscript for similar omissions.

line 82 awkward sentence, suggested change “in this turn, explaining why ASP32 is conserved. Gly would be…”

line 107 reference to “evolutionary unconserved positions” that were “enriched in ligand, DNA and protein interactions” is confusing given the idea that functional sites tend to be conserved. Perhaps including the concept of ortholog vs. paralog conservations would be helpful.

Figure 3 please include a more detailed description of the upset plot, including what the columns and rows represent.

Line 215-217. Please explain how you defined meaningful biological units. For example what are “preferred assemblies” . Do protein interactions include homodimers or are they limited to different chains?

Figure 6. I suppose I do not understand the difference between the color ranges and the height of the bars. Each are described as normalized Shenkin scores? Perhaps I am confused because the green and red are flipped? Also, “normalised” is spelled wrong.

Examples of the different binding modes might be informative. Is there something unique about the bound structures for the less observed convex or combination binding modes? Perhaps the mixed mode includes whole protein binding partners, while the others might be peptides?

Reviewer #2: The authors perform a comprehensive analysis of structural and functional features of Ankyrin proteins and their variants. The analysis identifies some novel trends regarding the features of missense mutations, and particularly, it identifies five positions that do not get mutated and have distinct structural features than other residues. The authors also discuss the relevance of the results for protein modeling and engineering.

The topic studied is important, the manuscript is well written, and the analysis is carefully designed. The data and scripts are available via github. I have the following minor comments and suggestions:

1) While the focus of the study is the analysis of the missense mutations, this part of the study is a bit lacking. In particular, we do not know if the missense mutations analyzed had any functional effect. Thus, it would be informative to further analyze the missense into likely neutral mutations, which would be common variants with no reported effect, as well as disease mutations or at least those that are predicted to have some kind of functional effect.

2) It would be interesting if the authors could comment on the overall flexibility of the Ankyrin domain, and if/how it affects the different regions and their interaction with binding partners.

Reviewer #3: This is a very comprehensive study of ankyrin-repeat proteins, a ubiquitous class of proteins in nature that are also of great interest as biotherapeutics, and hence the findings are important. I recommend publication with a point of clarification requested: One of the main findings of the sequence analysis is that there are three 'surfaces' on these proteins - in addition to the concave surface where ligand-interactions have been most frequently observed to date, there is the convex surface and also a ridge. Can the authors provide some clarification on their conclusions about the other two surfaces - are they saying that these might potentially be interaction interfaces on the basis of the observed sequence variations, and can they clarify how many of the proteins analysed are known to use these surfaces for binding?

**Have the authors made all data and (if applicable) computational code underlying the findings in their manuscript fully available?**

Reviewer #1: **No: **provide multiple sequence alignment with labeled sequences and lists of PDBs that were used for data analysis

Reviewer #2: Yes

Reviewer #3: Yes

PLOS authors have the option to publish the peer review history of their article (what does this mean?). If published, this will include your full peer review and any attached files.

Reviewer #1: No

Reviewer #2: No

Reviewer #3: No

Figure Files:

Data Requirements:

Reproducibility:

References:

---

## [Editor Report · Decision Letter 1]

10 Aug 2021

Dear Prof Barton,

We are pleased to inform you that your manuscript 'Ankyrin Repeats in Context with Human Population Variation' has been provisionally accepted for publication in PLOS Computational Biology.

Best regards,

Roland L. Dunbrack Jr., Ph.D.

Associate Editor

PLOS Computational Biology

Arne Elofsson

Deputy Editor

PLOS Computational Biology

---

## [Editor Report · Acceptance letter]

18 Aug 2021

PCOMPBIOL-D-21-01007R1 

Ankyrin Repeats in Context with Human Population Variation

Dear Dr Barton,

I am pleased to inform you that your manuscript has been formally accepted for publication in PLOS Computational Biology. Your manuscript is now with our production department and you will be notified of the publication date in due course.

With kind regards,

Zsofi Zombor
